# MiRNAs Expression Profiling of Bovine (*Bos taurus*) Testes and Effect of bta-miR-146b on Proliferation and Apoptosis in Bovine Male Germline Stem Cells

**DOI:** 10.3390/ijms21113846

**Published:** 2020-05-28

**Authors:** Yuan Gao, Fei Wu, Yaxuan Ren, Zihui Zhou, Ningbo Chen, Yongzhen Huang, Chuzhao Lei, Hong Chen, Ruihua Dang

**Affiliations:** Key Laboratory of Animal Genetics, Breeding and Reproduction of Shaanxi Province, College of Animal Science and Technology, Northwest A&F University, Yangling, Xianyang 712100, China; gaoyuan710@nwsuaf.edu.cn (Y.G.); wufei@nwafu.edu.cn (F.W.); 2018010734@nwafu.edu.cn (Y.R.); zihui96121@nwafu.edu.cn (Z.Z.); ningbochen@nwsuaf.edu.cn (N.C.); hyzsci@nwafu.edu.cn (Y.H.); leichuzhao1118@nwsuaf.edu.cn (C.L.); chenhong1955@nwsuaf.edu.cn (H.C.)

**Keywords:** cattle, miR-146b, spermatogenesis, miRNAs, RNA-Seq

## Abstract

Spermatogenesis is a complex biological process regulated by well-coordinated gene regulation, including MicroRNAs (miRNAs). miRNAs are endogenous non-coding ribonucleic acids (ncRNAs) that mainly regulate the gene expression at post-transcriptional levels. Several studies have reported miRNAs expression in bull sperm and the process of spermatogenic arrest in cattle and yak. However, studies for the identification of differential miRNA expression and its mechanisms during the developmental stages of testis still remain uncertain. In the current study, we comprehensively analyzed the expression of miRNA in bovine testes at neonatal (3 days after birth, *n* = 3) and mature (13 months, *n* = 3) stages by RNA-seq. Moreover, the role of bta-miR-146b was also investigated in regulating the proliferation and apoptosis of bovine male germline stem cells (mGSCs) followed by a series of experiments. A total of 652 miRNAs (566 known and 86 novel miRNAs) were identified, whereas 223 miRNAs were differentially expressed between the two stages. Moreover, an elevated expression level of bta-miR-146b was found in bovine testis among nine tissues, and the functional studies indicated that the overexpression of bta-miR-146b inhibited the proliferation of bovine mGSCs and promoted apoptosis. Conversely, regulation of bta-miR-146b inhibitor promoted bovine mGSCs proliferation. This study provides a basis for understanding the regulation roles of miRNAs in bovine testis development and spermatogenesis.

## 1. Introduction

In mammals, male gametes are produced in testis by spermatogenesis [1], which is the process to produce good-quality sperm; however, the reproductive performance is severely affected if this process is not well regulated. Moreover, the testis environment is an important condition for spermatogenesis, and healthy testes are essential for good reproductive ability. With aging, the genome of male gametes cells exhibit phase-specific gene expression patterns [2], especially from newborn to puberty. In addition to protein-coding messenger RNAs, many ncRNAs participate in regulation [3,4], including miRNAs.

MiRNAs are endogenous ncRNAs with 18–25 nt in length. These molecules regulate the gene expression at either transcriptional or post-transcriptional levels by RNA–RNA interaction [5]. MiRNAs repress gene expression by degrading mRNA or inhibiting translation via binding the 3′ UTR sequences of target mRNAs [6]. MiRNAs play important roles in many tissues and associated with important biological processes related to spermatogenesis and testis development [7,8]; for example, miRNAs regulate the renewal of spermatogonial stem cell (SSCs) [9] and androgen-dependent spermatogenic events [10].

Several studies demonstrated the importance of miRNAs as potential biomarkers in spermatogenesis and testis development and are required for primordial germ cell development and spermatogenesis [11]. A number of results have revealed that the differential expression miRNAs are associated with spermatogenesis in human, mice, pig, white yak, canine and sheep [12,13,14,15,16]. Moreover, various studies reported that miRNA regulates SSCs renewal, proliferation, differentiation and apoptosis. Several stage-specific miRNAs are coordinatively regulated in mouse spermatogenesis, such as miR-221 (SSC-specific), miR-203 (premeiotic-specific) and miR-34b-5p (meiotic-specific) [17]. MiRNA-20 and miRNA-106a regulate SSCs renewal by targeting *STAT3* and *Ccnd1* [9]. The CHD1L–miR-486–MMP2 regulatory axis has been identified in the regulation of SSC stemness and growth properties [18]. Additionally, miR-663a promotes the proliferation and DNA synthesis while suppresses the early apoptosis of human SSCs by targeting *NFIX* via *Cyclin A2*, *Cyclin B1* and *Cyclin E1* [19]. MiR-34c enhances mouse SSCs differentiation by targeting *Nanos2* [20]. As the studies on miRNAs regulating the spermatogenesis are emerging, it is crucial to understand the regulatory function of miRNA on spermatogenesis to identify the target molecules influencing male reproduction.

In bovine, several miRNAs were identified in male reproduction, which explored its differential expression in the process of spermatogenic arrest in cattleyak [21,22,23]. Moreover, some studies reported the expression of miRNA in bull sperm [24,25]. However, studies in the identification of differential expression of miRNA in different testis development stages and the exploration of miRNA mechanisms are still lacking. According to the developmental regularity of bovine testis described in our previous study [4], we chose three neonatal (three days after birth) and three adult (13 months) cattle testes to analyze the miRNA expression profiles combining with the characteristics and timing of testicular development [26]. Six RNA sequencing (RNA-Seq) libraries were constructed and sequenced. Then, we investigated the role of bta-miR-146b in regulating bovine male germline stem cells proliferation and apoptosis through a series of experimental verifications. This study aimed to identify key miRNAs related to testis development and spermatogenesis. Our data will provide a basis for improving breeding efficiency by predicting bull fertility.

## 2. Results

### 2.1. Small RNA Populations in Testes by Sequencing

Small RNA populations in the six Angus cattle testis libraries were analyzed by using Illumina deep sequencing method. Sequencing of the six libraries produced 113,031,969 original sequence reads. Moreover, a total of 13,355,974 to 23,256,012 clean reads were collected after removing the low-quality reads and adaptor sequences (Table 1). Finally, we obtained 13,227,405 to 23,046,455 reads with sizes ranging from 18 to 35 nt. The size distribution and abundance of the small RNAs were different in the two development stages (Figure 1a, Appendix A). These sequences were further mapped to the reference genome (*B. taurus UMD3.1*) using bowtie program with no mismatch allowed. A total of 12,471,299 to 20,624,480 sequences were perfectly mapped to the bovine reference genome (Table 1).

### 2.2. Identification of the Known and Novel miRNAs in Bovine Testes

The mapped sequences were compared with the miRBase v20.0 database to identify the known miRNAs in bovine testes. A total of 566 known miRNA were identified. The prediction of novel miRNAs was performed using miREvo and mirdeep2 tools employing the special hairpin-like structure of miRNA precursor from unannotated reads. A total of 86 novel miRNAs were identified from the unannotated reads. The miRNA information was shown in Appendix A. Moreover, 141 and 13 miRNAs only existed in a neonatal and mature stage, respectively, while 498 miRNAs were identified in both stages (Figure 1b).

### 2.3. Differentially Expressed miRNAs between Testis at Three Days and 13 Months

To gain an in-depth understanding of the potential roles of miRNAs in testis development, it is essential to detect the differentially expressed (DE) miRNAs. The miRNA expression levels were analyzed using edgeR software, and DE miRNAs were screened between neonatal and mature after the number of reads was normalized to transcripts per million (TPM). A volcano plot (Figure 1c) indicated the pattern of 223 DE miRNAs (202 known and 21 novel miRNAs) between neonatal and mature cattle, of which 112 were over-expressed (92 known and 20 novel miRNAs) and 111 were under-expressed (110 known and one novel miRNAs) in mature bovine testis when compared to the neonatal group (*p-adj* value < 0.05, |log2fold-change| > 1). Of these DE miRNAs, the top three over-expressed miRNAs were bta-miR-449a, bta-miR-34c and bta-miR-34b, with an increased fold-change of 9.41, 7.42 and 7.08, respectively; the top three under-expressed miRNAs were bta-miR-433, bta-miR-495 and bta-miR-196b, with a decreased fold-change of 3.99, 3.83 and 3.66, respectively. However, bta-miR-146b showed minimum *p-adj* value of all different expression miRNAs. The information of different expression miRNAs was shown in Appendix A.

### 2.4. Prediction of Target Genes for DE miRNAs and KEGG Enrichment Analysis

To identify the potential functional roles of these miRNAs, target gene prediction was performed using three software (miRanda, PITA and RNAhybrid). A total of 4664 miRNA-mRNA target pairs were predicted for the 124 DE known miRNAs and 18 DE novel miRNAs between neonatal and mature cattle were shown in Appendix A. The intersection of the target gene of DE miRNAs, which was predicted by three software packages, were used for the next Kyoto Encyclopedia of Genes and Genomes (KEGG) enrichment analysis. KEGG analysis identified nine pathways (*p* < 0.05), such as a Calcium signaling pathway, focal adhesion, oxytocin signaling pathway and cAMP signaling pathway. The top 20 enriched KEGG pathways are shown in Figure 2 and all results of the KEGG enrichment analysis are shown in Appendix A.

### 2.5. RT-qPCR Validation of DE miRNAs and Expression Level of bta-miR-146b in Bovine Tissues

To validate the expression level of DE miRNAs, we randomly chose the miRNAs that showed elevated expression and |log2fold-change|values. Stem-loop RT-qPCR was performed on seven known miRNAs (bta-miRNA-138, bta-miRNA-146b, bta-miRNA-34b, bta-miRNA-375, bta-miRNA-449a/b and bta-miRNA-495) and seven novel miRNAs (bta-novel-116, bta-novel-186, bta-novel-198, bta-novel-276, bta-novel-217, bta-novel-223 and bta-novel-427), whereas U6 was used as the internal reference. As shown in Figure 3a,b, a comparison of miRNA expression revealed that the expression of all the miRNAs selected was overexpressed in mature cattle compared with neonatal cattle except the downregulation of bta-miRNA-495, which was consistent with their expression patterns obtained from sequencing data.

Moreover, we found that the expression pattern of bta-miR-146b in RT-qPCR validation at two bovine testis development stages is consistent with small RNA-seq result, and its expression level and fold change are relatively high compared with other miRNAs. Therefore, the expression characteristic of bta-miR-146b in bovine tissues were detected. We found that the expression level of bta-miR-146b in fetal testis tissue was relatively higher than in other tissues (Figure 3c). Based on the above studies, we speculated that bta-miR-146b play a significant role in the regulation of bovine testis development. Therefore, we choose this miRNA to investigate its function in the proliferation and apoptosis of bovine mGSCs.

### 2.6. Bta-miR-146b Inhibits Bovine mGSCs Proliferation

To investigate the biological role of bta-miR-146b in bovine mGSCs proliferation, an exogenous bta-miR-146b mimics were transfected into cells to strengthen its expression level. After transfecting the bta-miR-146b mimic, the expression level of bta-miR-146b was significantly increased compared with the control group (*p* < 0.01) (Figure 4a). RT-qPCR showed that bta-miR-146b could decrease *PCNA*, *cyclinD1*, *p21* and *CDK2* mRNA level (Figure 4d). Additionally, western blot analysis indicated that bta-miR-146b could decrease the PCNA, CDK2 and cyclinD1 protein expression level (Figure 4c). Furthermore, CCK-8 analysis showed that bta-miR-146b group have a significantly decreased OD value (*p* < 0.01) compared with the control group (Figure 4b). Moreover, the number of EdU-positive cells in the bta-miR-146b group were reduced compared with the control group (Figure 4f). Furthermore, cell cycle assay by flow cytometry demonstrated that the G1 and G2 phase cell population was decreased and the S phase cell population was increased (Figure 4h). Next, we performed a functional loss experiment of bta-miR-146b. After the loss of bta-miR-146b function with its inhibitor, the mRNA expression level of *PCNA*, *CDK2*, *cyclinD1* and *p21* were significantly increased (Figure 4e) and the quantities of EdU-positive cells was increased (Figure 4g). Similarly, the S phase cell population was decreased and the number of cells in the G1 and G2 phase was increased (Figure 4i). Together, these results elucidated that bta-miR-146b could inhibit the proliferation in bovine mGSCs.

### 2.7. Bta-miR-146b Promotes Bovine mGSCs Apoptosis

To investigate the role of bta-miR-146b in bovine mGSCs apoptosis, we first detected some cell survival relative genes mRNA expression level, including *Bax*, *Bcl-2* and *Caspases 9*. Compared with the control group, overexpression of bta-miR-146b resulted in a significant increase of *Bcl-2*, *Bax* and *Caspases 9* expression (Figure 5a). Meanwhile, *Bax* and *Bcl-2* protein expression level was detected by western blot analysis. We found that protein levels were increased after overexpression of bta-miR-146b (Figure 5c). Therefore, we analyzed the ratio of *Bcl-2* and *Bax* mRNA expression level. The ratio was decreased after overexpression of bta-miR-146b (Figure 5b). Furthermore, cell apoptosis assay showed that the population of apoptosis cells was significantly increased compared in the bta-miR-146b group (28.2%) than that in the control group (13.0%) (Figure 5d). Above all, the results indicated that bta-miR-146b could promote bovine mGSCs apoptosis.

## 3. Discussion

With the increasing application of the artificial insemination, the fertility of one single bull affect thousands of female cattle. Selecting fertile males with superior genetic potency is essential to maintain the production and feeding of cattle. An abundant number of studies suggest that miRNA play an important role in testis development and spermatogenesis [9,10,11]. Therefore, this study aimed to explore crucial miRNAs in bull testis development and spermatogenesis as potential biomarkers for normal bull fertility used in predicting bull fertility.

In this study, a total of 652 miRNA (566 known and 86 novel miRNAs) were identified in bovine testes at two development stages by RNA-seq. Furthermore, 202 known and 21 novel miRNAs were differentially expressed between two stages. At the same time, most evidence suggests that numbers of miRNAs were related to bovine male reproduction. In 2011, 21 miRNAs were reported that were testis-specific miRNA compared with ovary [27]. Moreover, 11 miRNAs were observed in our study, among which bta-miR-1185, bta-miR-431, bta-miR-144, bta-miR-377, bta-miR-135b and bta-miR-654, which were underexpressed, while bta-miR-105a, bta-miR-449a/b/c and bta-miR-34b overexpressed in mature bovine testis when compared to the neonatal group. A total of 83 miRNAs were identified that were differentially expressed in High- and Low-motile sperm [28] and 26 miRNAs were differentially expressed in our study, as among which bta-let-7c/d/g, bta-miR-34b/c, bta-miR-6546. It is worth mentioning that the bta-miR-6546 was specifically expressed in cattle and may imply this miRNA was specifically regulating bovine male reproduction. Moreover, a number of miRNAs were found involved in the spermatogenic arrest in cattleyak [12,21,22]. These miRNAs, such as miR-34 [29], regulate spermatogenesis and perhaps embryogenesis, and miR-449, which regulates spermatogenesis, was also differentially expressed in our research. Another report suggests that transgenic constructs of miR-34b/c and miR-449a/b/c have shown that they are essential for normal spermatogenesis and male fertility, but their presence in sperm is dispensable for fertilization and preimplantation development [30].

MiRNA mainly performs its functions by binding to the target gene; thus, we predicted the target genes of DE miRNAs. A large number of DE miRNA target genes were predicted. To further explore the biological functions of miRNA in bovine testis, we performed KEGG pathway analysis using target genes of DE miRNAs. Through the analysis, various target genes were found enriched in KEGG pathways related to male reproduction, including calcium signaling pathway, focal adhesion, oxytocin signaling pathway, PI3K-Akt signaling pathway and MAPK signaling pathway. For example, *SLC25A31*(*solute carrier family 25 member 31*, also called *Ant4*) as the target gene of bta-miR-6525 was enriched in calcium signaling pathway. *Ant4* gene participates in male germ cell development in mice, and it is highly expressed during meiosis in adult mice [31]. *Ant4* null male mice exhibited infertile because of the meiotic arrest during leptotene stage [32]. *Nitric oxide synthase 3* (*NOS3*) as the target gene of bta-miR-2889 was enriched in calcium signaling pathway and oxytocin signaling pathway, which mutants might cause oxidative sperm DNA damage and increased risk of infertility in men [33,34]. The bta-miR-2387 target gene (*protein tyrosine kinase 2*, *PTK2*) in the focal adhesion pathway (also present in PI3K-Akt signaling pathway and chemokine signaling pathway) improves the movement of spermatids across the epithelium and preleptotene spermatocytes across the blood-testis barrier during spermatogenesis [35]. The encoded protein regulates actin polymerization and remodels actin cytoskeleton during acrosome reaction [36]. *Transforming growth factor beta 3* (*TGFB3*) was the target gene of bta-miR-365-5p present in MAPK signaling pathway, which has been demonstrated that activated by MAPK signals and effected blood-testis barrier (BTB) dynamics during spermatogenesis [37,38].

Bta-miR-146b was found to have the lowest *p*-value between two development stages and the highest expression in testis tissue. Therefore, we speculate bta-miR-146b may regulate bovine testis development and spermatogenesis; thus, we selected this miRNA for further verification. In this study, the immortalized bovine male germline stem cells (mGSCs) were chosen for functional study. Male germline stem cells, also named spermatogonial stem cells (SSCs), presented in the male testis, are responsible for spermatogenesis during their whole life [39,40]. As a result, we found that exogenous bta-miR-146b could decrease the expression level of *PCNA*, *CDK2*, *cyclinD1* and *p21* during bovine mGSCs growth stage. Meanwhile, the results of EdU, CCK-8 and flow cytometry assay also demonstrated that overexpression of bta-miR-146b impedes the proliferation of bovine mGSCs. However, the ratio of *Bcl-2* and *Bax* expression level was decreased. Moreover, the cell apoptosis assay showed that apoptosis cells were increased. Above all, bta-miR-146b inhibit proliferation and promote apoptosis of bovine mGSCs, while the opposite function in proliferation and apoptosis supports the theory that apoptosis is used to balance the effect of cell proliferation. Furthermore, miR-146 is expressed significantly high (180-fold) in undifferentiated spermatogonia than differentiating spermatogonia and it was suggested that miR-146 has involved in the control of retinoid acid-induced spermatogonial differentiation in the mouse [41]. The miR-146b expression level of the seminal plasma in patients with nonobstructive azoospermia was significantly increased compared with fertile controls [42], and the expression of miR-146 is lower in seminomas and spermatocytic seminomas compared with normal testes [43]. This suggests that miR-146b plays an important role in spermatogenesis.

## 4. Materials and Methods

### 4.1. Animal Samples Collection and RNA Isolation

All experimental design and procedures were performed by the Regulations for the Administration of Affairs Concerning Experimental Animals (Ministry of Science and Technology, Beijing, China, 2004). The study was approved by the guidelines of the ethics committee of the Northwest A&F University (Yangling, Shaanxi, China) (approval number: 20171208–010, 8 December 2017). 

Whole testes from six Angus bulls were collected from Shaanxi Kingbull Livestock Co., Ltd. (Baoji, China). The bulls are at two stages: pre-sex maturation (neonatal, 3 days after birth, *n* = 3) and post-sex maturation (mature, 13 months, *n* = 3). The heart, liver, spleen, lung, kidney, muscle, intestine, testis and stomach were collected from neonatal cattle (*n* = 3). All the individuals selected were healthy and unrelated. The detailed procedures of sample collection and RNA isolation refer to our previous studies [4].

### 4.2. Small RNA Sequencing and Analysis

A total of 3 μg RNA from each sample was used as input material for generation of the small RNA library. Small RNA libraries were generated using NEBNext^®^ Multiplex Small RNA Library Prep Set for Illumina^®^ (NEB, Ipswich, MA, USA.) and sequenced on an Illumina Hiseq 2500/2000 platform (Novogene, Beijing, China) and 50bp single-end reads were generated. Reads mapping were using Bowtie [44]. miRBase v20.0 database and modified software mirdeep2 [45] were used to identify known miRNAs. The software miREvo [46] and mirdeep2 were integrated to predict novel miRNA. DESeq R package (1.8.3) was used for differential expression analysis with TPM (transcript per million) criteria [47]. Predicting the target gene of miRNA was performed miRanda [48], PITA [49] and RNAhybrid [50]. KOBAS software [51] was used to analyze the statistical enrichment of the candidate target genes in KEGG pathways.

### 4.3. Cell Lines

Bovine male germline stem cells (mGSCs) were derived from an immortalized cell line [52], which was < P15 with male germline stem cell properties after immortalizing. The functional feature of the cell was checked by RT-PCR (Appendix A). The culture medium consisted of DMEM/F12 supplemented with 10% fetal bovine serum (Gibco, Carlsbad, CA, USA), 0.1 mM β-mercaptoethanol (Sigma-Aldrich, St. Louis, MO, USA), 2 mM L-glutamine (Invitrogen, Carlsbad, CA, USA), 100 U/mL penicillin and 100 mg/mL streptomycin (Invitrogen). Cells were seeded in 60-mm plate (2 × 10^6^ cells/well) with 4 mL medium in 5% CO_2_ at 37 °C.

### 4.4. cDNA Preparation and Quantitative Real-Time PCR

Total RNA purified and complementary DNA (cDNA) synthesis was performed by PrimeScript^TM^ RT reagent Kit with gDNA Eraser (Takara, Beijing, China). The RT-qPCR analyses were performed using the SYBR Green PCR Master Mix (Vazyme, Nanjing, China). The miRNA primers are listed in Appendix A and other primers are listed in Appendix A. The relative expression level of RT-qPCR data was calculated by the 2^−ΔΔCt^ method [53] with *β-actin* as an internal control. The experiment was biologically repeated three times and technically repeated three times for each group.

For the DE miRNAs expression validation, animal samples are the same in RNA-seq. Validated miRNAs were chosen among those with high differential expression (*p*-adj value < 0.03 and |log2fold-change| value > 1).

### 4.5. Western Blot Analysis, Cell Counting Kit-8 (CCK-8) and 5-Ethynyl-2-Deoxyuridine (EdU) Assay

Cells for the WB analysis were seeded in a six-well plate (4 × 10^5^ cells/well) with 2 mL culture medium and collected and extracted by total protein extraction Kit (Solarbio, Beijing, China). Total protein concentration was detected by using the BCA Protein Assay Kit (Solarbio, Beijing, China). The antibodies source information was listed in Table 2.

Cell proliferation was examined by using the CCK-8 (Multisciences, Hangzhou, China) and EdU assays (RiboBio, Guangzhou, China). The cells were plated into 96-well culture plates at a density of 5 × 10^3^ cells/well in 100 μL of culture medium/well and treated with miR-146b mimics or inhibitor. The procedures are in line with previous research [54].

### 4.6. Cell Cycle and Apoptosis Assay

We analyzed the cell cycle of different treatment groups using a cell cycle testing kit (Multisciences, Hangzhou, China). Cells were plated in 60-mm plates (2 × 10^6^ cells/well) with 4 mL culture medium treated with miR-146b mimics or inhibitor. The procedures were described in previous research [55].

### 4.7. Statistical Analysis

All the result analyses were performed with SPSS 19.0 (SPSS Inc., Chicago, IL, USA). The statistical significance was analyzed using a one-way analysis of variance. *p* < 0.05 was considered a statistically significant difference. Results were expressed as mean ± SEM; * *p* < 0.05, ** *p* < 0.01.

## 5. Conclusions

Our research comprehensively analyzed the miRNA expression profiles in bovine testes and this is the first time to investigate the function of bta-miR-146b in bovine mGSCs, which inhibit proliferation and promote apoptosis. Furthermore, our study provided an miRNA list that is a valuable resource for understanding their regulatory roles in bovine testis development and spermatogenesis. In the future, much effort should be exerted to investigate the role of individual miRNA involved in testis development and explore the pathways involved to reveal the mechanisms of bovine spermatogenesis.

## Figures and Tables

**Figure 1 ijms-21-03846-f001:**
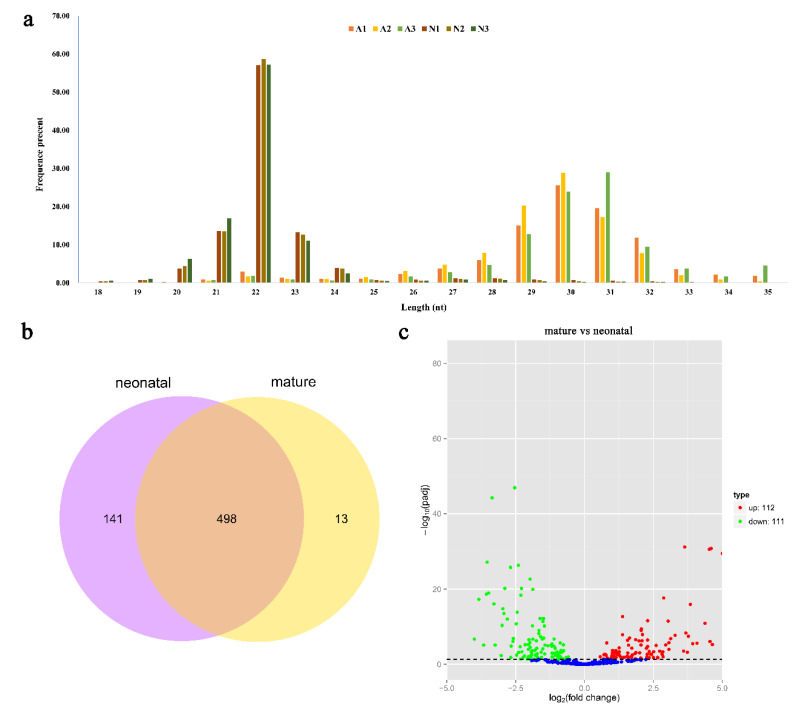
Sequencing and bioinformatics analysis results of miRNA in bovine testis. (**a**) The length distribution of small RNA library in six bovine testes. (**b**) Venn diagram showing overlap of miRNAs among neonatal and mature bovine testes. (**c**) Differential miRNA expression between neonatal and mature bovine testes. Volcano plots showing –log10 (*p-adj* value) versus log_2_ (fold change) difference in miRNA abundance in TPM (transcript per million) between neonatal and mature bovine testis. Red circles denote significantly overexpressed miRNAs, whereas green circles denote significantly underexpressed miRNAs in mature bovine testis when compared to the neonatal group. The cut off values (*p-adj* value < 0.05 and |log2fold-change| > 1) are selected for the differentially expressed (DE) analysis.

**Figure 2 ijms-21-03846-f002:**
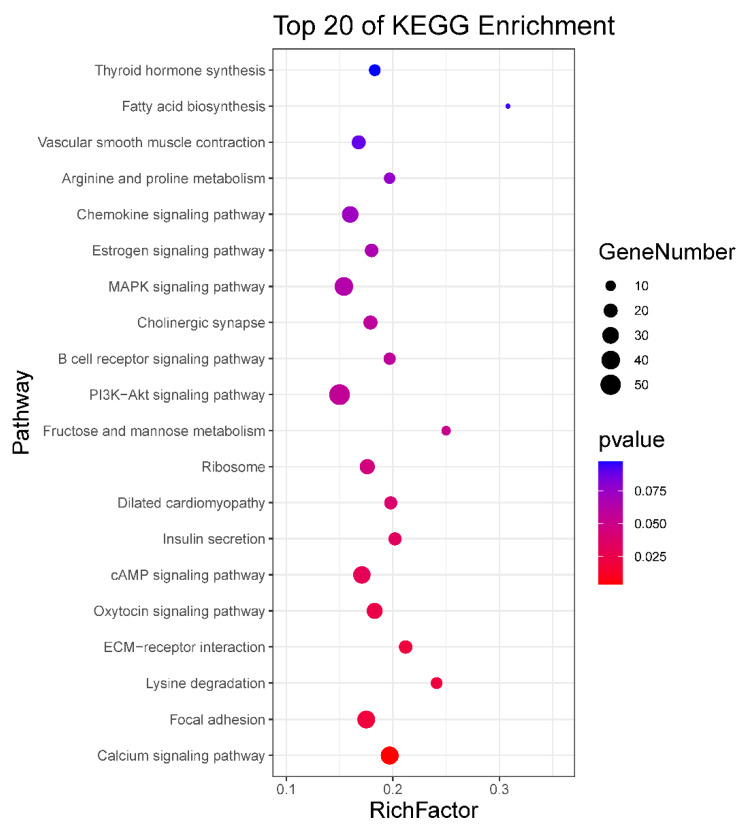
KEGG enrichment analysis for target gene of DE miRNAs. The top 20 enriched KEGG pathways are listed for target gene of DE miRNAs.

**Figure 3 ijms-21-03846-f003:**
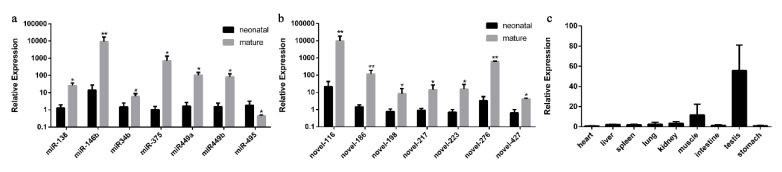
Validation of differentially expressed miRNAs by RT-qPCR. RT-qPCR validation of known (**a**) and novel (**b**) miRNA expression changes between neonatal and mature cattle testis. (**c**) The expression feature of bta-miR-146b in fetal tissues, including heart, liver, spleen, lung, kidney, muscle, intestine, testis and stomach. The data are present as the mean ± standard deviation of the mean; ** *p* < 0.01, * *p* < 0.05. The *Y*-axis represents the values that were calculated by log10.

**Figure 4 ijms-21-03846-f004:**
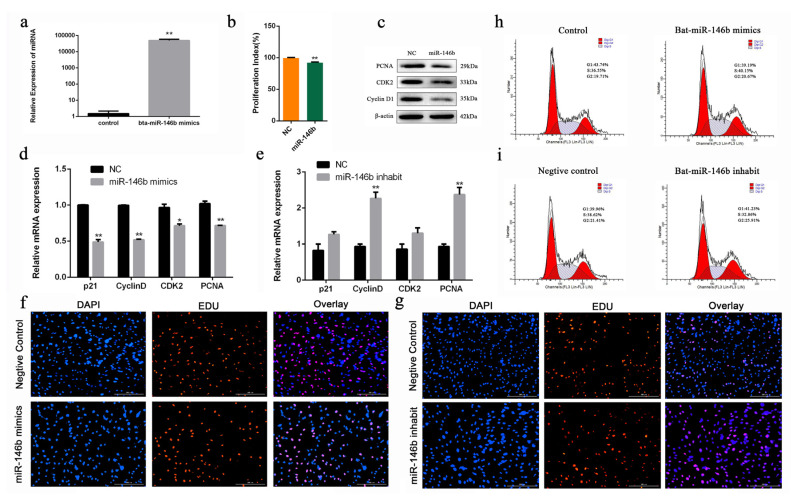
Bta-miR-146b regulates proliferation of bovine male germline stem cells. (**a**) The detection of bta-miR-146b expression efficiency after transfection of the bta-miR-146b expression plasmid by qRT-PCR. (**b**) Cell proliferation status was detected at 450 nm with CCK-8 reagent after overexpression of bta-miR-146b. (**c**) Cell cycle-related genes (*PCNA*, *CDK2* and *Cyclin D1*) protein expression level was detected by western blot analysis after overexpressing bta-miR-146b. (**d**) Cell cycle-related genes (*PCNA*, *CDK2*, *cyclinD1* and *p21*) mRNA expression level was detected by qRT-PCR after a gain or loss (**e**) of bta-miR-146b. *β-Actin* was used as the reference gene. (**f**) EdU staining-positive muscle cells was detected by EdU kit after a gain or loss (**g**) of bta-miR-146b. EdU (red), DAPI (blue); the scale bars represent 200 µm. (**h**) Distribution of cell numbers in each cell cycle stage was detected by PI flow cytometry after the gain or loss (**i**) of bta-miR-146b. Data are presented as mean ± SEM; * *p* < 0.05 and ** *p* < 0.01. DAPI: 4′,6-diamidino-2-phenylindole; EdU: 5-ethynyl-2-deoxyuridine; miR: microRNA; mRNA: messenger RNA; NC: negative control; PI: propidium iodide; qRT-PCR: quantitative real-time polymerase chain reaction.

**Figure 5 ijms-21-03846-f005:**
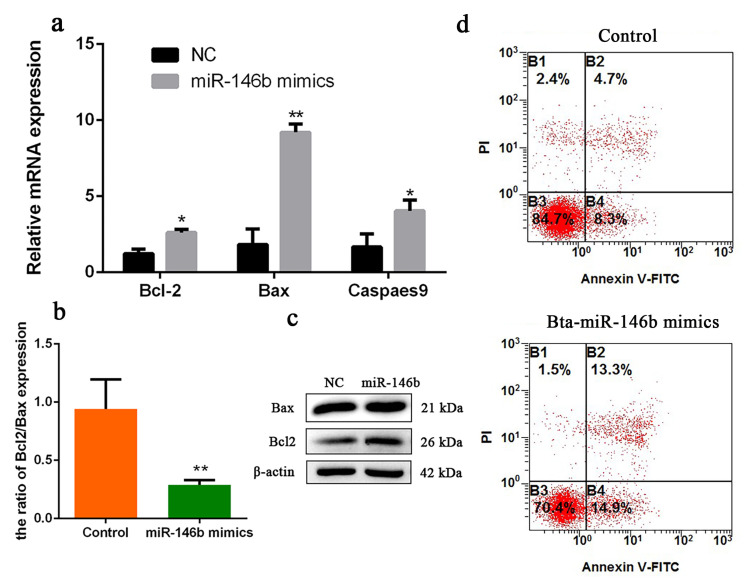
Bta-miR-146b regulates apoptosis of bovine male germline stem cells. (**a**) Cell apoptosis-related genes (*Bcl-2*, *Bax* and *Caspase 9*) mRNA expression levels were detected after overexpressing bta-miR-146b by qRT-PCR. (**b**) The ratio of *Bcl-2/Bax* gene expression after overexpressing bta-miR-146b. (**c**) Cell apoptosis-related genes (*Bcl-2* and *Bax*) protein expression level was detected by western blot analysis after a gain of bta-miR-146b. *β-Actin* was used as the reference gene. (**d**) Apoptotic muscle cells were detected by annexin V-FITC/PI-staining flow cytometry after a gain of bta-miR-146b. Data are presented as mean ± SEM; * *p* < 0.05 and ** *p* < 0.01. FITC: fluorescein isothiocyanate; miR: microRNA; mRNA: messenger RNA; NC: negative control: PI: propidium iodide; qRT-PCR: quantitative real-time polymerase chain reaction.

**Table 1 ijms-21-03846-t001:** Read statistics for six bovine testes miRNA-seq data.

Sample	Raw Reads	Bases	GC content	Clean Reads	Mapped Clean Reads
A1	21,735,749	1.087G	48.97%	21,217,578	20,716,723
A2	23,674,352	1.184G	48.25%	23,256,012	23,046,455
A3	20,884,976	1.044G	49.84%	20,620,423	20,341,478
N1	13,511,848	0.676G	48.76%	13,355,974	13,227,405
N2	15,612,572	0.781G	48.65%	15,424,567	15,269,334
N3	17,612,472	0.881G	48.86%	17,417,217	17,145,246

**Table 2 ijms-21-03846-t002:** The information of antibodies used for western blot.

Antibody	Dilution	Catalog Number	Brand
PCNA	1:1000	A0264	ABclonal
CyclinD1	1:1000	A11310	ABclonal
CDK2	1:1000	A18000	ABclonal
Bcl2	1:1000	A0208	ABclonal
Bax	1:500	A11550	ABclonal
β-actin	1:20000	AC038	ABclonal

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
