# Peer review of "MiRNAs Expression Profiling of Bovine (Bos taurus) Testes and Effect of bta-miR-146b on Proliferation and Apoptosis in Bovine Male Germline Stem Cells"

_ijms, 2020, doi:10.3390/ijms21113846_

Round 1

Reviewer 1 Report

The manuscript provides impportant data on miRNA expression profile in bovine testes together with molecular mechanizm of apoptosis and proliferation of germ cells. The manuscript is well-described and documented. Before publication additional information should be added to the text

  1. Give robust explanation on number of animals used for sequencing. refer to genetic differences (where there inbreed bulls).
  2. Provide information on cell line (used passage number and analyses that you use for checking biochemical and functional featutes of the cells)

Author Response

[Comment of Reviewer]

The manuscript provides important data on miRNA expression profile in bovine testes together with molecular mechanizm of apoptosis and proliferation of germ cells. The manuscript is well-described and documented. Before publication additional information should be added to the text

  1. Give robust explanation on number of animals used for sequencing. refer to genetic differences (where there inbreed bulls).

Reply: Thank the reviewers for these precious comments and suggestions. The testes were collected from 3 days after birth (n=3) and 13 months (n=3) cattle. All the individuals selected were healthy and unrelated. The details were revised in line 249-252.

  1. Provide information on cell line (used passage number and analyses that you use for checking biochemical and functional featutes of the cells).

Reply: We are grateful for the suggestion. Bovine mGSCs were derived from an immortalized cell line (Lei et al., 2017). In our study, the cell line was < P15 with male germline stem cell properties after immortalizing.  Lei et al., detected the SSC-specific markers (PLZF and LIN28A) and pluripotent markers (OCT4, NANOG, SOX2, and KLF4) were by RT-PCR. And SSC-specific markers (PLZF, CD49F, and PGP9.5), germ cell markers (VASA), and self-renewal markers (ETV5) were analyzed by immunofluorescence. Besides, the differentiation and proliferation potential of the immortalized bovine mGSCs were also examined. All the results showed that this cell line maintains the criteria of bovine mGSCs. Additionally, the other research has been analyzed the effect of LncRNA H19 and IGF-1 on the proliferation and apoptosis of bovine mGSCs by using this cell line. However, we detected the marker genes (OCT4, DAZL, PLZF,  and GFRA1) in the immortalized bovine mGSCs by RT-PCR before using in this study, the results are shown as follows:

Therefore, we confirmed that the cell line could be used in our study. The information of the cell line was added in line 266-268.

“Bovine male germline stem cells (mGSCs) were derived from an immortalized cell line [52], which was < P15 with male germline stem cell properties after immortalizing. The functional feature of the cell was checking by RT-PCR (Figure S1).”

Reviewer 2 Report

This study has potential and adding value to the research field. There are many aspect to improve before publication.

Title is misleading and does not reflect the outcome.

Why authors have used the testes after the birth of 3 days and 13 months, this needs to be explain and supported by.

The clear hypothesis and clinical relevance of the study is lacking, how does the role of miRNA regulation is related to spermatogenesis and development?

Line 118; typo software

I suggest to sketch out the over all experimental design and outcome

Figure 1a, the histogram of the distribution of miRNA was not clear, how that retrieved. What basis so the expression was used. The volcano plot is good to show the differential display but this should be first and then the DE can be visualized for the overlap analysis. What was the cut off values selected for the DE analysis.

Figure 2 Quality of the images not good. What was the basis for the selection of target miRNA for validation?

Figure 3. I recommend to use  some zoom in high magnification images, the message is not clear. Over all the quality of image is poor to comprehend the out come.

The primer information table can be placed in supplementary information.

Author Response

[Comment of Reviewer]

This study has potential and adding value to the research field. There are many aspect to improve before publication.

  1. Title is misleading and does not reflect the outcome.

Reply:  We are grateful for the suggestion. We have revised the title as “MiRNAs Expression Profiling of Bovine (Bos taurus) Testes and bta-miR-146b effects on Proliferation and Apoptosis in Bovine Male Germline Stem Cells”

  1. Why authors have used the testes after the birth of 3 days and 13 months, this needs to be explain and supported by.

Reply: We are grateful for the suggestion. Testis development includes the formation of testis tissue during embryonic stage and the growth of testis tissue at the postnatal stage. The testes of the bull grow relatively slowly until approximately 25 weeks of age and then a rapid phase of growth occurs until puberty, at 37–50 weeks of age (Rawlings et al., 2008). Lunstra et al. (1982) reported that Angus bulls reached first completed mating at 354 days. Considering the farm environment and the standards of body weight, we selected 13-month-old Angus bull as the sample of sexually mature cattle. In order to explore the process of sexual maturation in the bull calf, 3d after birth (neonatal, Pre-sex-maturation, n=3) and 13 months old (mature, Post-sex maturation, n=3) Angus bulls were selected. Additionally, the results of hematoxylin and eosin staining of bovine neonatal and mature testes were in our previous study, which also provided the gist for choosing these two stages. These basis that described in our previous study were as the reference in this study in line 68-69.

  1. The clear hypothesis and clinical relevance of the study is lacking, how does the role of miRNA regulation is related to spermatogenesis and development?

Reply: Thank you for your precious suggestion. We have made large modifications in the introduction section to clarify the miRNA regulation related to spermatogenesis and development, as well as our experimental hypothesis and purpose. The revised part was in line 40-76.

  1. Line 118; typo software

Reply: Sorry for typing error. The wrong word has been substituted.

  1. I suggest to sketch out the overall experimental design and outcome

Reply: Thank you for your suggestion. The figure that sketch out the overall experimental design and outcome was showed as follows:

  1. Figure 1a, the histogram of the distribution of miRNA was not clear, how that retrieved. What basis so the expression was used. The volcano plot is good to show the differential display but this should be first and then the DE can be visualized for the overlap analysis. What was the cut off values selected for the DE analysis.

Reply: Thank you for your suggestion. We have improved all images quality, the changed figure was shown as follows:

We also submit all figures as the individual file in the submission system. The histogram is sketched by the proportion of miRNA length distribution (The ratio of the number of miRNA of a certain length to the total miRNA). We try to draw the histogram using the number of miRNAs, but some data were not shown because of the big difference of miRNA number in different length. Therefore, we choose the proportion to show the length distribution. Meanwhile, we submit the excel of the proportion data of length distribution as supplement material.

Besides, the Venn of the overlap analysis is mainly exhibited the number of stage-specific miRNA in two development stage based on the read counts. Then, the expression of miRNA was calculated. Therefore, we put Venn in front of volcano plot followed the analysis steps.

The cut-off values selected for the DE analysis are P-value < 0.05 and  |log2fold-change|>1. These are mentioned in line 104-105, and we have added the values in figure caption (line 482 – line 483).

  1. Figure 2 Quality of the images not good. What was the basis for the selection of target miRNA for validation?

Reply: Thanks. The quality of the images was improved. MiRNAs were randomly chosen with high expression and |log2fold-change| values. And animal samples for the DE miRNAs expression validation are the same in RNA-seq. We added these details in Material and Methods (line 281 – line 282). The revised figure was shown as follows:

  1. Figure 3. I recommend to use some zoom in high magnification images, the message is not clear. Over all the quality of image is poor to comprehend the out come.

Reply: Thank you for your advice. Since the quality of images added in the word will be compressed, we submitted the manuscript as well as the individual figures. In order to make the quality of the image more clearer, we improved the resolution of all figure from 300dpi to 600dpi. The changed image is shown as follows:

  1. The primer information table can be placed in supplementary information.

Reply: Thank you for your suggestion. The primer information table has moved to the supplementary material (Table S2 and S3).

Reviewer 3 Report

This manuscript describing miRNA profiles in early life and mature young bulls and role of miRNA 146 on spermatogenesis contains interesting and new information which deserve to be published. However, the manuscript presents many important flaws and corresponding issues need to be addressed. Critical information is missing in the material and methods section. The analysis of part the results is incomplete and the corresponding interpretation superficial. The results are insufficiently discussed. The quality / definition of the figures is poor and important details are hard to see. In addition, the English is extremely poor. Almost every sentence contains language mistakes which makes corresponding parts very difficult to understand. In many places, it is difficult to know really what the authors mean. There is an absolute need for the manuscript to be edited and revised by a person for whom English is the native language and to plan common work to make the meaning of the sentences not ambiguous.

Detailed comments

Lines 35-37: The sentence should be clarified. Which kind of biological processes have been addressed from previous work.

Methods

Lines 225-229: The description of the origin of biological material is not clear. Are the testis issued from 6 different bulls.

Lines 272-273: “293T cell lines were provided by our laboratory”. The description should provide more details and also cite a reference, if the production of these cells has been described previously.

Lines 274-279: Cells were used after how many passages. A few details are given later, however, the type of flask / volume of media used and number of cells should be mentioned here.

Lines 281-285: Which genes were used for normalization. The number of replicates performed should be mentioned in this part of the text, not only in the figure legend lines 470-471.

RNA was purified from how many cells and what was the amount obtained.

Lines 287-288: Please mention the amount of cells used.

Lines 298-299 and 308-309: Indicate number of passages before use

Results:

Line 80: The sub-title should be “Results” and not “Results and Discussion” and there is a specific part for discussion.

Line 102 and everywhere else in the text when needed: Abbreviations should be explicated at first occurrence even as for “DE” the meaning is obvious.

Lines 108-110 and later line 112: The way the comparison was made is not clear. “112 were up regulated and 111 were down regulated” when considering which stage … same when reading figure legends. Is it mature minus immature or the inverse ? it is possible to understand later on, but things should be clear from start in this part of text.

Also, the terms “up and down regulated” should be replaced systematically (in all text) and respectively by “over and under expressed” as results are purely descriptive and there is, at this stage of work, no explanation and no relevant information to speak about “regulation”.

Line 112 and line 114 : Sentence should be clarified as “…. Increased fold change of 9.41, 7,42 and 7.08 respectively..” no so many decimals should be labelled.

Line 114: “…. and 3.66 “.

Line 119: “Total score” and “total energy” should be explained and defined in Material and Methods section.

Lines 120-121: This part of the results is really too short and the analysis is incomplete. The reader is sent/should refer to illustrations without any guidance. A pathway analysis should be performed and a list of the most important genes belonging to pathway(s) possibly influenced by the differentially expressed miRNA between the two stages provided.

Lines 122-126: The choice, criteria used to select the miRNAs for the validation study should be explained before in Material and Methods. The chosen ones correspond to / address critical genes (groups of genes) or where they chosen at random ? The kind of cells used for validation should be mentioned here also.

Lines 131-132: “…the expression is “consistent” with bovine testis development. The term “consistent” refers to discussion…. It’s use here in the result section without reference is ambiguous. Does the authors mean instead that the expression increases with age / is higher in the mature bulls.

Lines 140 and the following … until line 167 : P values should be indicated in text for this series of results.

Discussion:

Lines 172-175 are just a repeat of the objectives and may not be needed.

Line 183: “…. When compared to varies “ should be added by the end of the sentence. Again, the term over expressed should be used instead of “up-regulated”.

Lines 184-196: This part of discussion is relatively poor and should be “enriched” form the above suggested analyses. Genes and pathways targeted by the differentially expressed miRNA should be mentioned and the results discussed with literature … at least how these genes are possibly involved in testis development /what is their possible role in this process and how the differences observed in miRNAs between stages are consistent with the above.

Lines 207-209: Add reference(s)

Lines 325-333: Conclusion should probably follow the discussion.

Line 464: The sentence is not clear without knowing the way the comparison was done (see comment above, lines 108-110).

Author Response

[Comment of Reviewer]

This manuscript describing miRNA profiles in early life and mature young bulls and role of miRNA 146 on spermatogenesis contains interesting and new information which deserve to be published. However, the manuscript presents many important flaws and corresponding issues need to be addressed. Critical information is missing in the material and methods section. The analysis of part the results is incomplete and the corresponding interpretation superficial. The results are insufficiently discussed. The quality / definition of the figures is poor and important details are hard to see. In addition, the English is extremely poor. Almost every sentence contains language mistakes which makes corresponding parts very difficult to understand. In many places, it is difficult to know really what the authors mean. There is an absolute need for the manuscript to be edited and revised by a person for whom English is the native language and to plan common work to make the meaning of the sentences not ambiguous.

Reply:  thank you for your precious advice. We have added the information that the detail comments mentioned in the material and methods, result and discussion section. In order to make the quality of the image more clearer, we improved the resolution of all figure from 300dpi to 600dpi. Since the quality of images added in the word will be compressed, we submitted the manuscript as well as the individual figures. And the English level of the manuscript has been improved by the native language speaker.

Detailed comments

  1. Lines 35-37: The sentence should be clarified. Which kind of biological processes have been addressed from previous work.

Reply : Thank you for your suggestion. This sentence has been modified to “MiRNAs were found that play important roles in many tissues and are associated with kinds of important biological processes related to spermatogenesis and testis development [7,8], for example, miRNAs as regulators of spermatogonial stem cell (SSCs) renewal and androgen-dependent spermatogenic events.” We have added examples in line 45 - line 46.

  1. Lines 225-229: The description of the origin of biological material is not clear. Are the testis issued from 6 different bulls.

Reply : Sorry for our unclearly meaning. The 6 testes used for sequencing and RT-qPCR are collected from 6 different bulls. This sentence has been revised in line 248-250.

 “Whole testes from six Angus bulls were collected from Shaanxi Kingbull Livestock Co., Ltd. (Baoji, China). The bulls are at two stages: pre-sex maturation (neonatal, 3 days after birth, n=3) and post-sex maturation (mature, 13 months, n=3).”

  1. Lines 272-273: “293T cell lines were provided by our laboratory”. The description should provide more details and also cite a reference, if the production of these cells has been described previously.

Reply : Sorry for our carelessness. Due to multiple revision of the article version, 293T was not used in this study, so this sentence has been deleted.

  1. Lines 274-279: Cells were used after how many passages. A few details are given later, however, the type of flask / volume of media used and number of cells should be mentioned here.

Reply : Thanks for your suggestion. In our study, the mGSCs cell line was < P15 with male germline stem cell properties after immortalizing. The type of flask, volume of media and number of cells have been mentioned in line 272.

“Cells were seeded in 60-mm plate (2 x 106 cells/well) with 4 mL medium in 5% CO2 at 37°C.”

  1. Lines 281-285: Which genes were used for normalization. The number of replicates performed should be mentioned in this part of the text, not only in the figure legend.

Reply: Thanks. β-actin was used for normalization and the number of replicates performed has been moved to this part of the text from figure legend. The revised sentence was shown in line 278 – line 280.

“The relative expression level of RT-qPCR data were calculated by the 2−ΔΔCt method [53] with β-actin as an internal control. The experiment was biologically repeated three times and technically repeated three times for each group.”

  1. lines 470-471. RNA was purified from how many cells and what was the amount obtained.

Reply: Thanks. The RNAs used for the DE miRNAs expression validation and tissue expression profile are from the animal tissue samples. Total RNAs were extracted from testis tissue using Trizol reagent. And the RNA purified and cDNA synthesis was performed by PrimeScriptTM RT reagent Kit with gDNA Eraser (Takara, Beijing, China). According to the manufacturer’s instruction, the amount of RNA used in the experiment can be calculated according to the concentration of RNA. The procedures of DE miRNAs expression validation were revised in 274-282.

“5.4. cDNA preparation and quantitative real-time PCR

Total RNA purified and complementary DNA (cDNA) synthesis was performed by PrimeScriptTM RT reagent Kit with gDNA Eraser (Takara, Beijing, China). The RT‐qPCR analyses were performed using the SYBR Green PCR Master Mix (Vazyme, Nanjing, China). The miRNA primers are listed in Table S6 and other primers are listed in Table S7. The relative expression level of RT-qPCR data was calculated by the 2−ΔΔCt method [53] with β-actin as an internal control. The experiment was biologically repeated three times and technically repeated three times for each group.

For the DE miRNAs expression validation, animal samples are the same in RNA-seq. MiRNAs were randomly chosen with high expression and |log2fold-change| values.”

  1. Lines 287-288: Please mention the amount of cells used.

Reply: Thanks. The details of cell culture for WB were add in line 284.

“Cells for the WB analysis were seeded in 6-well tissue culture plate (4 x 105 cells/well) with 2 mL culture medium”

  1. Lines 298-299 and 308-309: Indicate number of passages before use

ReplyThanks. the number of passages was described in line 267.

“Bovine male germline stem cells (mGSCs) were derived from an immortalized cell line [52], which was < P15 with male germline stem cell properties after immortalizing.”

  1. Line 80: The sub-title should be “Results” and not “Results and Discussion” and there is a specific part for discussion.

Reply: Thanks. The sub-title has been revised.

  1. Line 102 and everywhere else in the text when needed: Abbreviations should be explicated at first occurrence even as for “DE” the meaning is obvious.

Reply: Thanks. The abbreviations have been explicated in the whole manuscript.

  1. Lines 108-110 and later line 112: The way the comparison was made is not clear. “112 were up regulated and 111 were down regulated” when considering which stage … same when reading figure legends. Is it mature minus immature or the inverse ? it is possible to understand later on, but things should be clear from start in this part of text.

Also, the terms “up and down regulated” should be replaced systematically (in all text) and respectively by “over and under expressed” as results are purely descriptive and there is, at this stage of work, no explanation and no relevant information to speak about “regulation”.

Reply: Thank you for your precious advice. In this section, the neonatal group was used as the control group. The sentence has been revised as follows “……112 were overexpressed (92 known and 20 novel miRNAs) and 111 were underexpressed (110 known and 1 novel miRNAs) in mature bovine testis when compared to the neonatal group (P-value < 0.05, |log2fold-change|>1).” Also, we used “over and under expressed” to place “up and down regulated” in all text, the revised words were in red color.

  1. Line 112 and line 114 : Sentence should be clarified as “…. Increased fold change of 9.41, 7,42 and 7.08 respectively..” no so many decimals should be labelled.

Line 114: “…. and 3.66 “.

Reply: Thanks. The sentence has been clarified in line 106-108.

  1. Line 119: “Total score” and “total energy” should be explained and defined in Material and Methods section.

Lines 120-121: This part of the results is really too short and the analysis is incomplete. The reader is sent/should refer to illustrations without any guidance. A pathway analysis should be performed and a list of the most important genes belonging to pathway(s) possibly influenced by the differentially expressed miRNA between the two stages provided.

Reply: Thank you for your suggestion. This part has been revised as follows:

Result:

2.4. Prediction of target genes for DE miRNAs and KEGG enrichment analysis

To identify the potential functional roles of these miRNAs, target gene prediction was performed using three software (miRanda, PITA, and RNAhybrid). A total of 4664 miRNA-mRNA target pairs were predicted for the 124 DE known miRNAs and 18 DE novel miRNAs between neonatal and mature cattle were shown in Table S4. The intersection of target gene of DE miRNAs, which predicted by three software, were used for the next Kyoto Encyclopedia of Genes and Genomes (KEGG) enrichment analysis. KEGG analysis identified 9 pathways (P < 0.05), such as Calcium signaling pathway, Focal adhesion, Oxytocin signaling pathway, and cAMP signaling pathway. The top 20 enriched KEGG pathways were shown in Figure 2 and the all result of KEGG enrichment analysis were shown in Table S5.

Materials and Methods:

Predicting the target gene of miRNA was performed miRanda [38],PITA [49], and RNAhybrid [50]. KOBAS software [51] was used to analyze the statistical enrichment of the candidate target genes in KEGG pathways.”

In the revised text, in order to improve the reliability of the results, three prediction software was used to predict miRNA target genes. The parameters of the software are the default. “Total score” and “total energy” are not used for cutoff values.

  1. Lines 122-126: The choice, criteria used to select the miRNAs for the validation study should be explained before in Material and Methods. The chosen ones correspond to / address critical genes (groups of genes) or where they chosen at random ? The kind of cells used for validation should be mentioned here also.

Reply: Thank you for your suggestion. MiRNAs were randomly chosen with high expression and |log2fold-change| values. And animal samples for the DE miRNAs expression validation are the same in RNA-seq. We add these details in Material and Methods (line 281 – line 282).

“For the DE miRNAs expression validation, animal samples are the same in RNA-seq. MiRNAs were randomly chosen with high expression and |log2fold-change| values.”

  1. Lines 131-132: “…the expression is “consistent” with bovine testis development. The term “consistent” refers to discussion…. It’s use here in the result section without reference is ambiguous. Does the authors mean instead that the expression increases with age / is higher in the mature bulls.

Reply: Thank you for your suggestion. In this part, we aimed to validate the authenticity of  RNA-seq data by RT-qPCR, and screened reliable miRNAs for later experiments. Therefore, the “consistent” means that the miRNA expression pattern examined in RT-qPCR are the same with RNA-seq data.

  1. Lines 140 and the following … until line 167 : P values should be indicated in text for this series of results.

Reply: Thanks. A part of P values is indicated in text for this series of results in red color. However, the P values of genes are different in RT-qPCR analysis. The text will be wordy when all the P value described in the text part, and they were clearly in Figure 4d/e and 5a. Moreover, the figure caption and statistical analysis in the Materials and Methods section were in a good description of the P value represented by *and **.

Discussion:

  1. Lines 172-175 are just a repeat of the objectives and may not be needed.

Reply: Thanks. Line 172-175 has been deleted.

  1. Line 183: “…. When compared to varies “ should be added by the end of the sentence. Again, the term over expressed should be used instead of “up-regulated”.

Reply: Thanks. This sentence has been revised as follows:

“ 11 miRNAs were observed in our study, in which, bta-miR-1185, bta-miR-431, bta-miR-144, bta-miR-377, bta-miR-135b, and bta-miR-654 were underexpressed and bta-miR-105a, bta-miR-449a/b/c and bta-miR-34b were overexpressed in mature bovine testis when compared to the neonatal group.”

  1. Lines 184-196: This part of discussion is relatively poor and should be “enriched” form the above suggested analyses. Genes and pathways targeted by the differentially expressed miRNA should be mentioned and the results discussed with literature … at least how these genes are possibly involved in testis development /what is their possible role in this process and how the differences observed in miRNAs between stages are consistent with the above.

Reply: We are grateful for your precious suggestion. We have added some discussion about the target genes miRNA and enrich KEGG pathway in this part.

“MiRNA mainly performs its functions by binding to the target gene, so we predicted the target genes of DE miRNAs. A large number of DE miRNA target genes were predicted by three software. And to further explore the biological functions of miRNA in bovine testis, we performed KEGG pathway analysis using target genes of DE miRNAs. Through the analysis, various target genes were found enriched in KEGG pathways related to male reproduction, including calcium signaling pathway, focal adhesion, oxytocin signaling pathway, PI3K-Akt signaling pathway, and MAPK signaling pathway. For example, SLC25A31(solute carrier family 25 member 31, also called Ant4) as the target gene of bta-miR-6525 was enriched in calcium signaling pathway. Ant4 gene participates in male germ cell development in mice, and it highly expressed during meiosis in adult mice [31]. Ant4 null male mice exhibited infertile because of the meiotic arrest during leptotene stage [32]. Nitric oxide synthase 3 (NOS3) as the target gene of bta-miR-2889 was enriched in calcium signaling pathway and oxytocin signaling pathway, which mutants might cause oxidative sperm DNA damage and increased risk of infertility in men [33,34]. The bta-miR-2387 target gene (protein tyrosine kinase 2, PTK2) in the focal adhesion pathway (also present in PI3K-Akt signaling pathway and chemokine signaling pathway) improves the movement of spermatids across the epithelium and preleptotene spermatocytes across the blood-testis barrier during spermatogenesis [35]. The encoded protein regulates actin polymerization and remodels actin cytoskeleton during acrosome reaction [36]. Transforming growth factor beta 3 (TGFB3) was the target gene of bta-miR-365-5p present in MAPK signaling pathway, which has been demonstrated that activated by MAPK signals and effected blood-testis barrier (BTB) dynamics during spermatogenesis [37,38].”

  1. Lines 207-209: Add reference(s)

Reply: Thanks. The reference has been added.

  1. Lines 325-333: Conclusion should probably follow the discussion.

Reply: Thanks. The Conclusion section has been moved to follow the Discussion.

  1. Line 464: The sentence is not clear without knowing the way the comparison was done (see comment above, lines 108-110).

Reply: We are grateful for the suggestion. The sentence has been revised as follows:

“Red circles denote significantly overexpressed miRNAs, whereas green circles denote significantly underexpressed miRNAs in mature bovine testis when compared to the neonatal group.”

Round 2

Reviewer 1 Report

The manuscript was corrected according suggestions.

Author Response

Thank the reviewers for these precious comments and suggestions.

Reviewer 2 Report

Authors answered the queries 

Author Response

Thank the reviewer for these precious comments and suggestions.

Reviewer 3 Report

Most of the comments have been well adressed by the authors and the manuscript has been very much improved.

In the introduction, lines 70-71. It is not fully clear how this work would contribute directly to select high fertility bulls. This would eventually find it's place in the end of dicsussion, but in that case, the authors should clarify the context and explain how the information form this work could be used in the future. 

Line 317: The sentence "MiRNAs were randomly chosen from those highly expressed..." is somewhat confusing because then these  MiRNA were not chosen at random ... The sentence should be replaced by "Validated miRNAs were chosen among those with high differential expression (adj P value < XXX ? ) and with > XXX ? l log2 fold changes l values."

This type of choice for validation is not be the best as only the most significant ones are validated, increasing the likelyhood of getting similar changes .....It would preferable to have some other miRNA "more borderline"/ close to Padj = 0.05 validated also and such an information would reinforce better the validity of the results. ... 

Author Response

Most of the comments have been well addressed by the authors and the manuscript has been very much improved.

  1. In the introduction, lines 70-71. It is not fully clear how this work would contribute directly to select high fertility bulls. This would eventually find its place in the end of discussion, but in that case, the authors should clarify the context and explain how the information form this work could be used in the future.

Reply: Thanks for the precious suggestions. With the thoughtful consideration, we think that “select high fertility bull” is not an accurate way of describing our results. So we change the this word to “by predicting bull fertility” in line 71. And we have added the explanation in discussion (line 210-215).

“With the increasing application of the artificial insemination, the fertility of one single bull affect thousands of female cattle. Selecting fertile males with superior genetic potency is essential to maintain the production and feeding of cattle. An abundance of researches suggested that miRNA play an important role in testis development and spermatogenesis [9,10,11]. Therefore, this study aimed to explore crucial miRNAs in bull testis development and spermatogenesis as potential biomarkers for normal bull fertility used in predicting bull fertility.”

  1. Line 317: The sentence "MiRNAs were randomly chosen from those highly expressed..." is somewhat confusing because then these MiRNA were not chosen at random ... The sentence should be replaced by "Validated miRNAs were chosen among those with high differential expression (adj P value < XXX ? ) and with > XXX ? l log2 fold changes l values."

Reply: We are grateful for the suggestion. We have change this sentence in line 322-324.

“Validated miRNAs were chosen among those with high differential expression (P-adj value < 0.03 and |log2fold-change| value > 1).”

  1. This type of choice for validation is not be the best as only the most significant ones are validated, increasing the likelyhood of getting similar changes .....It would preferable to have some other miRNA "more borderline"/ close to Padj = 0.05 validated also and such an information would reinforce better the validity of the results. ...

Reply: Thanks for the precious advice. In this part, we aimed to verify the reliability of the RNA-seq data and select crucial miRNA in testis development and spermatogenesis for next research. Since the differences of the calculation method of miRNA expression between bioinformatic analysis and RT-qPCR, the expression value of miRNA have a certain deviation. Moreover, in our results, the foldchange of the miRNAs which P-adj values are close to 0.05 are small

We thought that these miRNAs were not considered as the candidate miRNAs, and the verification of 14/223 DE miRNAs can prove our RNA-seq results. Therefore, we hold the opinion that the verification of miRNA close to P-adj = 0.05 is not necessary.